# DMSO Improves the Ski-Slope Effect in Direct PCR

**Joo-Young Kim** [1,*] **, Ju Yeon Jung** [1] **, Da-Hye Kim** [1] **, Seohyun Moon** [1] **, Won-Hae Lee** [1] **, Byung-Won Chun** [2] **and Dong-Ho Choi** [2]

1   Forensic DNA Division, National Forensic Service, Wonju 26460, Korea; jjy7@korea.kr (J.Y.J.); snoopy8627@gmail.com (D.-H.K.); vetman97@gmail.com (S.M.); onesun@korea.kr (W.-H.L.)
2   DNA Analysis Division, National Forensic Service, Busan Institute, Busan 50612, Korea; warney@korea.kr (B.-W.C.); pihodong@korea.kr (D.-H.C)
*   Correspondence: jykim7112@korea.kr; Tel.: +82-33-902-5715

**Featured Application: This paper's application of the proposed novel method is reducing the ski-slope effect in direct PCR, thus aiding more efficient amplification of DNA obtained from crime scenes.**

**Abstract:** Analytical techniques such as DNA profiling are widely used in various fields, including forensic science, and novel technologies such as direct polymerase chain reaction (PCR) amplification are continuously being developed in order to acquire DNA profiles efficiently. However, non-specific amplification may occur depending on the quality of the crime scene evidence and amplification methods employed. In particular, the ski-slope effect observed in direct PCR amplification has led to inaccurate interpretations of the DNA profile results. In this study, we aimed to reduce the ski-slope effect by using dimethyl sulfoxide (DMSO) in direct PCR. We confirmed that DMSO (3.75%, *v/v*) increased the amplification yield of large-sized DNA sequences more than that of small-sized ones. Using 50 Korean buccal samples, we further demonstrated that DMSO reduced the ski-slope effect in direct PCR. These results suggest that the experimental method developed in this study is suitable for direct PCR and may help to successfully obtain DNA profiles from various types of evidence at crime scenes.

**Keywords:** direct PCR; short tandem repeat; ski-slope effect; DMSO; forensic DNA

## 1. Introduction

First developed by Alec Jeffreys in 1985, DNA fingerprinting has evolved rapidly since the 1990s and found applications in domains such as criminal investigations, individual identification, and paternity tests [1–3]. The development of scientific investigation techniques leading to a high success rate of DNA extraction from human tissue samples—including crime scene samples—and improvements in short tandem repeat (STR) amplification have further resulted in more reliable DNA profiling results. In addition, more efficient direct polymerase chain reaction (PCR) amplification techniques for obtaining DNA profiles from samples have been recently developed [4–7]. The direct PCR method, known as colony PCR, has been widely used for rapid detection of gene cloning and transfection in the field of microbiology since the early 1990s [8,9]. This method directly amplifies DNA from samples and has the advantage of minimizing DNA loss during the extraction process. Van Oorschot et al. [10] reported that 20–70% of sample DNA was lost during the DNA extraction process, with a possibility of contamination with foreign DNA. Direct PCR circumvents intermediate stages such as DNA extraction and quantification, making it a rapid and labor-effective method for successfully obtaining DNA profiles from samples [11,12]. This advantage has led to the continuous development of the direct PCR method since the mid-2000s and has recently been applied to contact evidence from crime scenes [4,13–15]. Additionally, Jung et al. [16] reported a rapid and cost-effective

modification of direct PCR involving the use of direct PCR buffer in reference buccal samples. However, since direct PCR amplifies the target gene without DNA extraction, purification, or quantification from the sample, it can lead to non-specific amplification and cause stutters, ski-slope effect, split peaks, allele imbalance, and allele drop-in/out due to the state of the sample and various PCR inhibitors [17,18]. For this reason, continuing research resulting in refinement of techniques to successfully obtain DNA profiles through direct PCR is warranted.

Even though PCR is one of the most important techniques to obtain a DNA profile, the poor condition of the samples to be amplified can lead to abnormal or insufficient amplification of the target sequences. Thus, many studies have focused on these aspects for further improvement [19]. Particularly, various organic additives such as dimethyl sulfoxide (DMSO) have been reported to improve the amplification yield and specificity of PCR [19–22] and are referred to as PCR additives or enhancers. In this study, we confirmed the applicability of the additive DMSO in reducing the ski-slope effect, which is an issue arising due to non-specific amplification in direct PCR, observed especially when using Prep-n-Go buffer. We aimed to develop an improved experimental method for direct PCR that can be applied in forensics to successfully obtain DNA profiles from various types of evidence at crime scenes.

## 2. Materials and Methods

### 2.1. Reagents

Standard 2800M Control DNA (Promega, WI, USA) was used for the experiment. DMSO (1–5%, *v/v*) was purchased from Sigma-Aldrich (St. Louis, MO, USA). Prep-n-Go Buffer, GlobalFiler PCR Amplification Kit, and Quantifiler Trio DNA Quantification Kit were purchased from Applied Biosystems (Carlsbad, CA, USA).

### 2.2. Sample Collection

Collection of 50 Korean buccal samples was performed as described previously by Jung et al. [16] using Oral Cell sampling kit (OC card; Cat. No.: TNT-SP-001; TNT Research, Jeonju, Korea). To produce the punch sample containing identical number of cells, saliva diluted in distilled water was deposited on the OC card, and the center of the spot was evenly punched with a diameter of 1.2 mm using BSD600-Duet Automated Dried Sample Punch Instrument (BDS Robotics, Brisbane, Queensland, Australia). Written informed consent was obtained from all study participants. Use of the samples and the analytical procedures involved were approved by the Institutional Review Board of the National Forensic Service of the Republic of Korea.

### 2.3. PCR Amplification and Direct PCR Amplification

PCR amplification of 2800M DNA was performed with dose-dependent concentrations of DMSO (1–5%, *v/v*) using the GlobalFiler PCR Amplification kit according to the manufacturer's instructions on a GeneAmp PCR system 9700 (Applied Biosystems). Briefly, 2 µg 2800M DNA, 2.5 µL Primer Set, and each concentration of DMSO were mixed with 7.5 µL Master Mix in a total reaction volume of 25 µL. The PCR conditions were as follows: initial denaturation at 95 °C for 1 min, 29 cycles of denaturation at 94 °C for 10 s, annealing and extension at 59 °C for 90 s, and final extension at 60 °C for 10 min.

Direct PCR amplification was performed using Prep-n-Go buffer—a direct PCR reagent—according to a previously established modified method [16]. Omitting the pretreatment step, we directly added the Prep-n-Go buffer to the GlobalFiler PCR Amplification Master Mix, followed by amplification using GeneAmp PCR system 9700. Briefly, 2 µg 2800M DNA or one 1.2 mm-punched saliva sample, 2.5 µL Primer Set, 0.9 µL DMSO (3.75%, *v/v*), and 2 µL Prep-n-Go buffer were mixed with 7.5 µL Master Mix in a total reaction volume of 25 µL. The amplified PCR product was electrophoresed on 4% agarose gel and visualized with Image Analyzer (UVP GelSolo, Analytik Jena, Jena, Germany).

Intensity of the amplified PCR product was measured by Image J software (open-source program, https://imagej.nih.gov, accessed on 4 December 2020).

### 2.4. Direct PCR Quantification

The DNA amplified by direct PCR was quantified using the Quantifiler Trio DNA Quantification Kit according to the user manual on a 7500 Real-Time PCR system (Applied Biosystems). The PCR reaction contained 10 μL Reaction Mix, 8 μL Primer Mix, 2 μg 2800M DNA or one 1.2 mm-punched saliva sample, 0.7 μL DMSO (3.75%, *v/v*), 1.6 μL Prep-n-Go buffer, and sterilized distilled water adjusted to a final volume of 20 μL in a 96-well optical reaction plate. The reaction conditions were as follows: 95 °C for 10 min, followed by 40 cycles of 95 °C for 15 s and 60 °C for 1 min. The fluorescence data were analyzed using the 7500 system SDS v1.2.3 (Applied Biosystems).

### 2.5. Capillary Electrophoresis and Analysis

For each sample, the PCR amplification product was mixed with 18.5 μL of Hi-Di Formamide and 0.5 μL of standard and heated to 95 °C for 3 min for denaturation. The denatured sample was stabilized at 4 °C and was subjected to capillary electrophoresis using 3500xL Genetic Analyzer (Applied Biosystems) according to the manufacturer's instructions. The final results were analyzed with GeneMapper ID-X v1.4 (Applied Biosystems). Analysis of the ski-slope effect according to the STR locus was performed using the relative fluorescence units (RFU) ratio calculated for each locus with the GeneMapper ID-X Software. Y-indel and DYS391 were excluded for calculations related to peak height.

## 3. Results and Discussion

### 3.1. Enhancement of PCR Amplification by DMSO in Multiplex PCR Amplification Kit

The effect of DMSO on multiplex PCR amplification was analyzed using the STR genotyping kit, GlobalFiler kit. In order to determine the appropriate concentration of DMSO for the multiplex PCR system, we first examined the amplification yield of 2800M DNA with dose-dependent concentrations of DMSO (0, 1, 2.5, 3.75, and 5%, *v/v*) with the GlobalFiler kit. The amplification yield of 2800M DNA was observed to increase in a dose-dependent manner at high DMSO concentrations, and it was most effective at a concentration of 3.75% (Figure 1A). Particularly, DMSO markedly improved the PCR amplification of large-sized DNA sequences (>200 bp) among the total sequences of 75 bp to 444 bp in the GlobalFiler kit, whereas amplification of DNA sequences (smaller than 200 bp) was slightly decreased (Figure 1B). This effect can play an important role in improving the ski-slope effect. Although Jung et al. [16] demonstrated high performance of their rapid technique for direct PCR using GlobalFiler kit, the issue of ski-slope effect affecting the intra-color peak balance could not be resolved. Therefore, we next examined if DMSO could alleviate the ski-slope effect in direct PCR system.

### 3.2. Reducing of Ski-Slope Effect by DMSO in Direct PCR

We investigated the potential of DMSO in the reduction of ski-slope effect and enhancement of amplification in direct PCR system using Prep-n-Go buffer. We observed that DMSO effectively improved amplification of 2800M DNA and 1.2 mm-punched saliva samples (Figure 2). In addition, even in direct PCR systems, DMSO improved the amplification of large-sized DNA sequences but not small-sized DNA sequences, as shown in Figure 1.

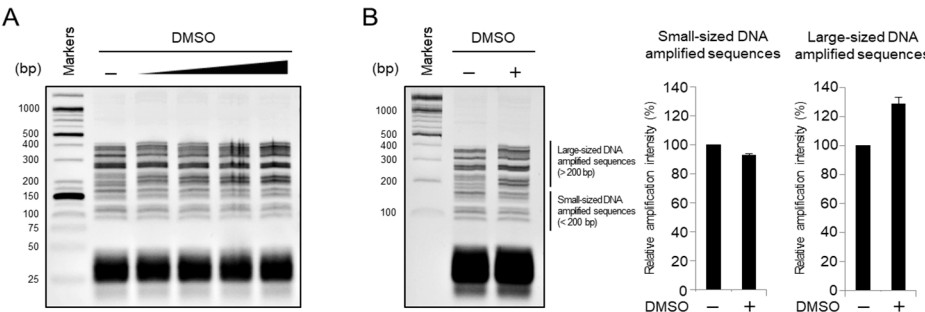

**Figure 1.** Effect of dimethyl sulfoxide (DMSO) on multiplex PCR amplification kit. (**A**) Amplification of 2800M DNA (2 μg) with dose-dependent concentrations of DMSO (0, 1, 2.5, 3.75, and 5%, *v/v*) using the GlobalFiler Kit. (**B**) Amplification of 2800M DNA (2 μg) with appropriate concentration of DMSO (3.75%, *v/v*) using the GlobalFiler kit. Amplification yield of 2800M DNA was compared in each of small- and large-sized DNA amplified sequences. Intensity of amplified PCR product was measured by Image J software.

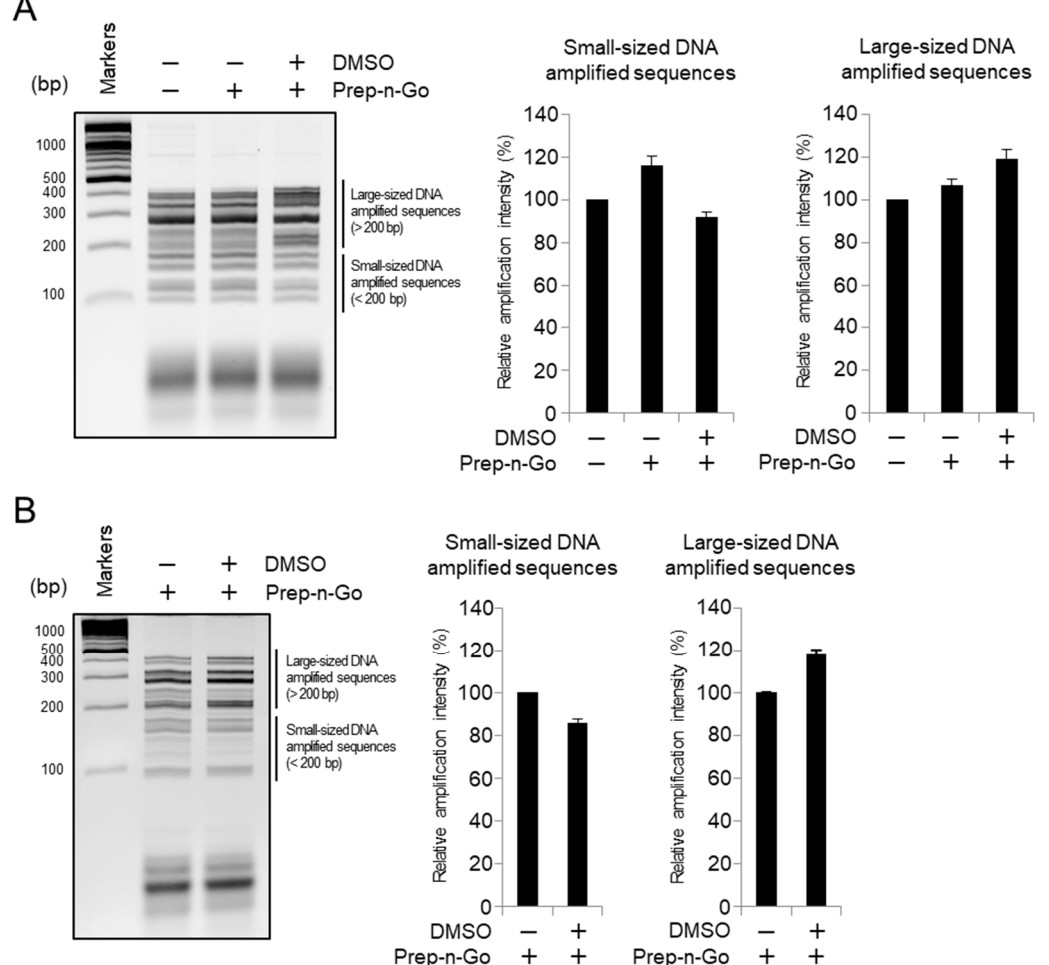

**Figure 2.** Effect of dimethyl sulfoxide (DMSO) in direct PCR performed using the GlobalFiler kit with Prep-n-Go buffer (2 μL/25 μL PCR reaction volume). Amplification yield with DMSO examined in 2800M DNA (**A**) and 1.2 mm-punched saliva samples (**B**) and compared in each of small- and large-sized DNA amplified sequences. Intensity of amplified PCR product was measured by Image J software.

To further identify these amplification patterns, we quantified the DNA amplified in the direct PCR system using large (amplicon size, 214 bp) and small (amplicon size, 80 bp) autosomal DNA fragments targeted by the Quantifiler Trio DNA Quantification Kit. These sequences can be used as indicators of DNA degradation by comparing the ratio of their quantification results [23]. We confirmed that the amplification of large autosomal DNA was reduced by Prep-n-Go buffer in direct PCR (Figure 3C), which in turn is associated with the ski-slope effect. However, we found that DMSO alleviated this issue, even in 1.2 mm-punched saliva samples (Figure 3C,D). Moreover, we confirmed that the amplification of small autosomal DNA was increased by Prep-n-Go buffer in direct PCR but slightly decreased by DMSO treatment (Figure 3A,B).

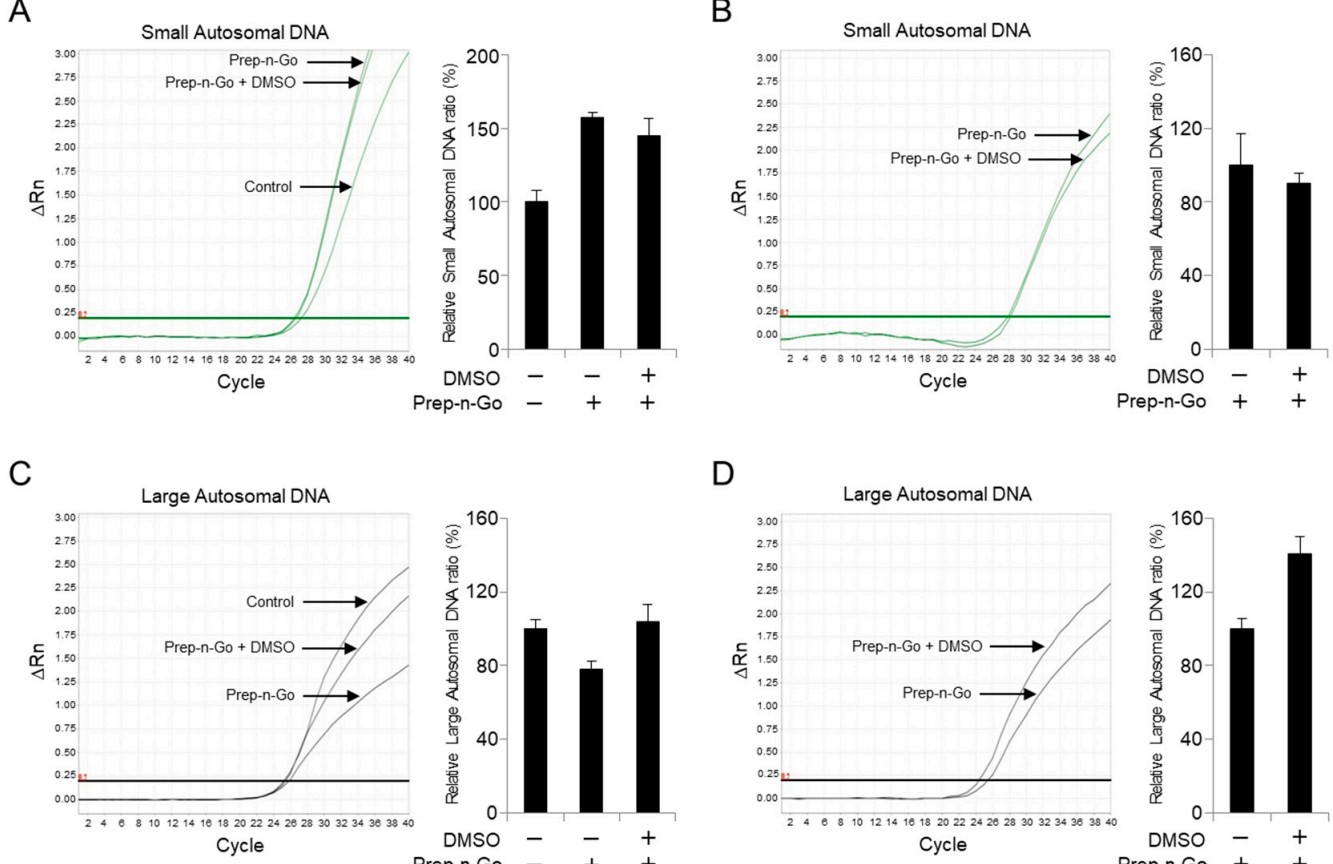

**Figure 3.** Quantitative analysis of small or large autosomal DNA by dimethyl sulfoxide (DMSO) amplified using direct polymerase chain reaction (PCR). Quantitative analysis of the amplified DNA was performed using small or large autosomal DNA (amplicon size: 80 or 214 bp, respectively) included in the Quantifiler Trio DNA Quantification Kit in 2800M DNA (**A,C**) and 1.2 mm-punched saliva samples (**B,D**). Data represent mean ± standard deviation.

Based on the results of PCR amplification enhanced by DMSO in direct PCR, we next examined whether DMSO could reduce the ski-slope effect in direct PCR with 50 Korean buccal samples using the OC card. In the STR genotyping kit, the ski-slope effect is generally shown as a decreasing aspect according to larger DNA amplification sizes of STR locus on the same color channel, such as blue dye (6-FAM (90.5~384.5 bp): D3S1358, vWA, D16S539, CSF1PO, and TPOX), green dye (VIC (108.5~347.5 bp): D8S1179, D21S11, and D18S51), yellow dye (NED (75.0~380.0 bp): D2S441, D19S433, TH01, and FGA), red dye (TAZ (83.5~444.0 bp): D22S1045, D5S818, D13S317, D7S820, and SE33), and purple dye (SID (80.0~355.5 bp): D10S1248, D1S1656, D12S391, and D2S1338) [24]. We observed that direct PCR with Prep-n-Go buffer alone led to a decrease in the RFU ratio of STR genotyping according to a large-sized STR locus in each color channel (Figure 4). However,

we confirmed that the RFU ratio was improved by addition of DMSO along with Prep-n-Go buffer in direct PCR. In particularly, DMSO more significantly improved the RFU ratio in STR locus of large-sized DNA sequences than in small-sized DNA sequences. For each color channel, DMSO enhanced the RFU ratio of STR genotyping by −5.4~+101.8% for blue dye (6-FAM), −5.5~+38.0% for green dye (VIC), −3.6~+42.6% for yellow dye (NED), −5.9~+34.8% for red dye (TAZ), and −3.0~+74.7% for purple dye (SID), as shown in Table 1. These results indicate that DMSO could be used to rectify the ski-slope effect in direct PCR through increased amplification of large-sized DNA sequences and reduced amplification of small-sized DNA sequences.

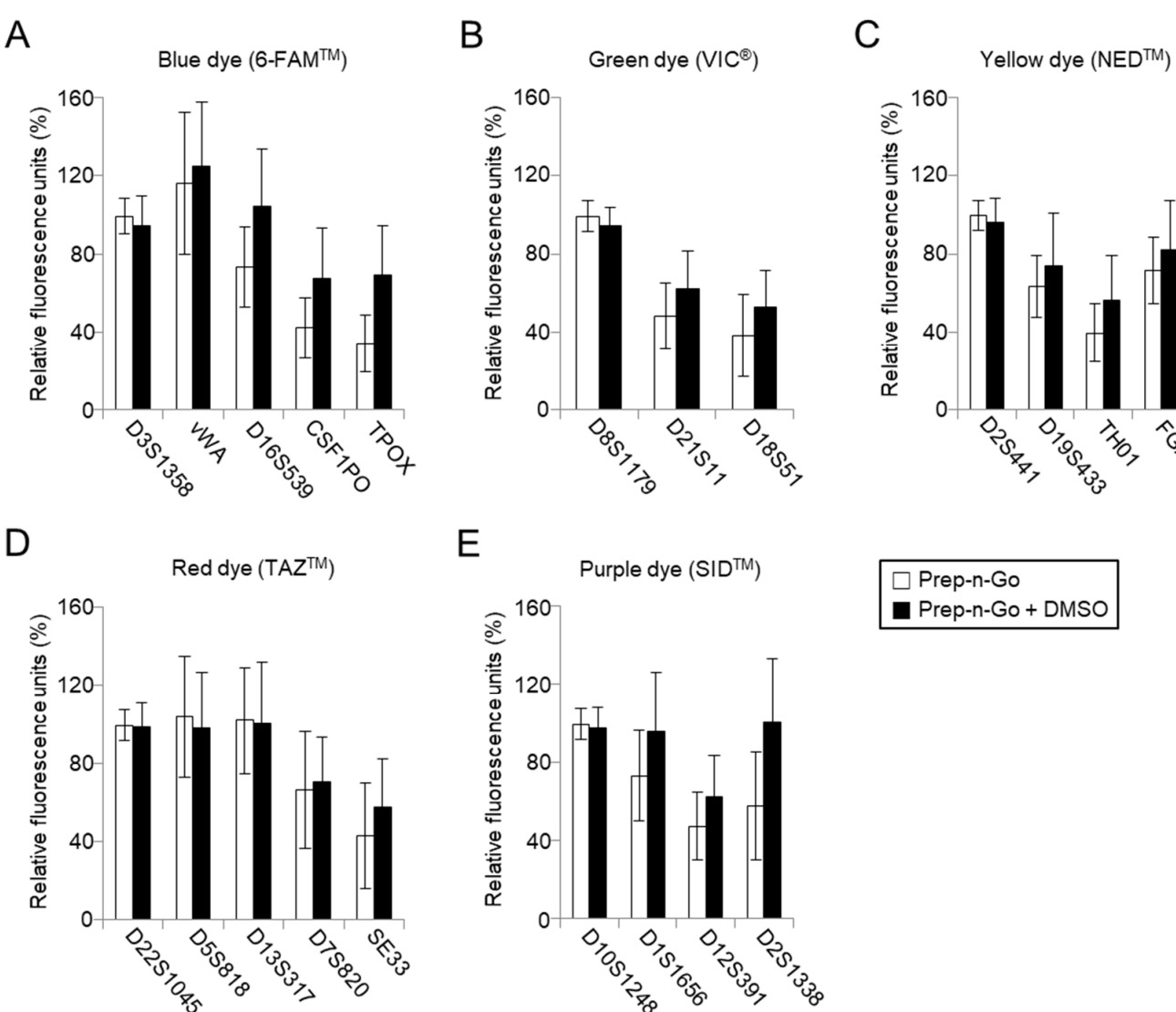

**Figure 4.** Reduction of ski-slope effect by dimethyl sulfoxide (DMSO) in direct polymerase chain reaction (PCR). The average relative fluorescence unit (RFU) ratio of short tandem repeat (STR) locus in each color channel was analyzed with the GlobalFiler kit using 50 Korean buccal OC card samples (**A–E**). Y-indel and DYS391 were excluded for calculation related to peak height. Data represent mean ± standard deviation.

**Table 1.** Dimethyl sulfoxide (DMSO) improved the relative fluorescence unit (RFU) ratio of short tandem repeat (STR) genotyping in each color channel.

| Dye | Locus | DNA Amplification Sizes (bp) | Average of Relative Fluorescence Units (%) | | Ratio of Change |
|---|---|---|---|---|---|
| | | | Prep-n-Go (*n* = 50) | Prep-n-Go + DMSO (*n* = 50) | |
| Blue dye (6-FAM) | D3S1358 | 90.5~146.5 | 100.0 ± 9.0 | 94.6 ± 15.1 | 5.4% ▼ |
| | vWA | 151.0~215.0 | 116.1 ± 36.2 | 124.7 ± 32.9 | 7.4% ▲ |
| | D16S539 | 221.5~273.5 | 73.5 ± 20.4 | 104.1 ± 29.2 | 41.7% ▲ |
| | CSF1PO | 277.0~325.0 | 42.2 ± 15.4 | 67.5 ± 25.6 | 59.9% ▲ |
| | TPOX | 332.5~384.5 | 34.3 ± 14.5 | 69.3 ± 25.0 | 101.8% ▲ |
| Green dye (VIC) | D8S1179 | 108.5~176.5 | 100.0 ± 8.1 | 94.5 ± 9.0 | 5.5% ▼ |
| | D21S11 | 179.5~246.5 | 48.2 ± 16.7 | 62.3 ± 19.2 | 29.2% ▲ |
| | D18S51 | 255.5~347.5 | 38.5 ± 21.1 | 53.1 ± 18.8 | 38.0% ▲ |
| Yellow dye (NED) | D2S441 | 75.0~113.5 | 100.0 ± 7.6 | 96.4 ± 12.3 | 3.6% ▼ |
| | D19S433 | 115.5~173.5 | 63.6 ± 15.8 | 73.8 ± 26.9 | 16.0% ▲ |
| | TH01 | 174.0~219.5 | 39.5 ± 14.9 | 56.3 ± 22.6 | 42.6% ▲ |
| | FGA | 221.0~380.0 | 71.4 ± 17.0 | 82.2 ± 24.8 | 15.1% ▲ |
| Red dye (TAZ) | D22S1045 | 83.5~126.5 | 100.0 ± 8.1 | 98.2 ± 12.4 | 1.8% ▼ |
| | D5S818 | 133.5~189.5 | 103.8 ± 31.1 | 97.7 ± 28.6 | 5.9% ▼ |
| | D13S317 | 197.0~249.0 | 101.8 ± 27.3 | 100.5 ± 31.1 | 1.3% ▼ |
| | D7S820 | 256.5~304.5 | 66.6 ± 29.8 | 70.6 ± 22.9 | 6.0% ▲ |
| | SE33 | 306.0~444.0 | 42.8 ± 26.8 | 57.7 ± 24.4 | 34.8% ▲ |
| Purple dye (SID) | D10S1248 | 80.0~132.0 | 100.0 ± 8.1 | 97.0 ± 10.6 | 3.0% ▼ |
| | D1S1656 | 154.0~209.5 | 72.9 ± 23.2 | 95.3 ± 30.4 | 30.7% ▲ |
| | D12S391 | 211.0~270.5 | 47.1 ± 17.3 | 62.5 ± 20.5 | 32.6% ▲ |
| | D2S338 | 275.5~355.5 | 57.5 ± 27.7 | 100.5 ± 32.5 | 74.7% ▲ |

## 4. Conclusions

Direct PCR system amplifies DNA directly from evidence samples and has the advantage of minimizing DNA loss during extraction, purification, or quantification processes. However, this advantage can lead to both non-specific amplification and the ski-slope effect. For this reason, a continuous improvement in direct PCR technique is required to successfully obtain DNA profiles.

In this study, we developed a novel method employing the addition of DMSO in direct PCR that resulted in markedly increased amplification; this may result in accurate interpretation of DNA profiles. DMSO (3.75%, *v/v*) more markedly increased the PCR amplification yield of large-sized DNA sequences (>200 bp) than small-sized DNA sequences (<200 bp) in direct PCR. In addition, DMSO increased the RFU ratio of large-sized locus in STR genotyping and consequently reduced the ski-slope effect in direct PCR using 50 Korean buccal samples. In addition to reducing the ski-slope effect, this novel method for direct PCR can aid in more efficient amplification of DNA samples acquired from various pieces of crime scene evidence.

**Author Contributions:** Conceptualization, J.-Y.K. and J.Y.J.; methodology, J.-Y.K. and D.-H.K.; analysis and validation, J.-Y.K., J.Y.J., and D.-H.K.; writing and original draft preparation, J.-Y.K.; writing—review and editing, S.M., W.-H.L., B.-W.C., and D.-H.C. All authors have read and agreed to the published version of the manuscript.

**Funding:** This work was supported by National Forensic Service (NFS2018DNA02 and NFS2020 DNA02), Ministry of the Interior and Safety, Republic of Korea.

**Institutional Review Board Statement:** The use of the samples and the analytical procedures involved were approved by the Institutional Review Board of the National Forensic Service of the Republic of Korea.

**Informed Consent Statement:** Written informed consent was obtained from all the study participants.

**Data Availability Statement:** The data presented in this study are available on request from the corresponding author.

**Conflicts of Interest:** The authors declare no conflict of interest.

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
