# Peer review of "DMSO Improves the Ski-Slope Effect in Direct PCR"

_applsci, doi:10.3390/app11041943_

Round 1

Reviewer 1 Report

The article is interesting for forensic caseworks in genetics and can be of help for those who work with degraded biological samples which is the case of forensic genetics. However, some issues should be improved or clarified.

  • Information contained in the paragraph 187-194 is redundant since I think it is already included in Table 1. In my view, this makes reading more difficult.
  • Table 1 contains key information although table 1 is not even mentioned within the text.
  • It seems that DMSO diminishes in fact ski-slope effect. However, it is not that clear there is a direct and univocal reduction. For example, table 1 shows that the reduction is not the same across dyes. Red-dye shows ratios of change lower than expected for size-ranges that are, however, larger in other dyes. Some discussion on this should be included. Perhaps, this is only an statistical effect but there are also other possibilities.
  • An issue that should be addressed is: Could the 50 swab samples be equally (and correctly) genotyped with the 3 categories studied (nothing, Prep-n-Go, Prep-n-Go+DMSO)? Was there a significant improvement?
  • In my opinion, although DMSO can be of help, a crucial experiment should test DMSO in true forensic degraded samples. Did the authors try this even just once?
  • Line 110. The authors should explain why they choose 40 cycles. This amount is not common at all for Global Filer. I presume they try to see better the sigmoid-curves shown in figure 3. Is this so?
  • In my opinion, supplementary material is scarce and can be easily included in the main text.

Reviewer 2 Report

"Improve/ment of ski-slope effect" should be replaced by e.g. reducing/diminishing of ski-slope effect.

English is not good in comparative sequences.

Description of large-sized and small-sized target fragments is too intricate.
